# Cytotoxicity and Lipase Inhibition of Essential Oils from Amazon Annonaceae Species

**André de Lima Barros** [1,*], **Emilly J. S. P. de Lima** [1], **Jéssica V. Faria** [1,2], **Leonard R. D. Acho** [3], **Emerson S. Lima** [3], **Daniel P. Bezerra** [4], **Elzalina R. Soares** [2], **Bruna R. de Lima** [5,6], **Emmanoel V. Costa** [5], **Maria Lúcia B. Pinheiro** [5], **Giovana A. Bataglion** [5], **Felipe M. A. da Silva** [5,6], **Nállaret M. Dávila Cardozo** [7], **José F. C. Gonçalves** [8] and **Hector H. F. Koolen** [1,*]

1   Grupo de Pesquisas em Metabolômica e Espectrometria de Massas, Escola Superior de Ciências da Saúde, Universidade do Estado do Amazonas, Manaus 690065-130, Brazil
2   Centro de Estudos Superiores de Tefé, Universidade do Estado do Amazonas, Tefé 69553-100, Brazil
3   Faculdade de Farmácia, Universidade Federal do Amazonas, Manaus 69077-000, Brazil
4   Instituto Gonçalo Moniz, Fundação Oswaldo Cruz, Salvador 40296-710, Brazil
5   Departamento de Química, Universidade Federal do Amazonas, Manaus 69080-900, Brazil
6   Centro de Apoio Multidisciplinar, Universidade Federal do Amazonas, Manaus 69077-000, Brazil
7   Instituto de Investigaciones de la Amazonía Peruana, Iquitos 784, Peru
8   Laboratório de Fisiologia e Bioquímica Vegetal, Instituto Nacional de Pesquisas da Amazônia, Manaus 69060-062, Brazil
*   Correspondence: andrelima1701@gmail.com (A.d.L.B.); hkoolen@uea.edu.br (H.H.F.K.)

**Abstract:** Essential oils from Amazonian species are gaining increasing interest worldwide due to their medicinal and cosmetic applications; however, the relation among the chemical constituents and their biological properties are not well explored. Therefore, the present research aims to obtain an understanding of the bioactivity of chemical compounds in the essential oils of plants from the Annonaceae family (*Bocageopsis pleiosperma*, *Onychopetalum amazonicum*, *Unonopsis duckei*, *U. floribunda*, *U. rufescens*, *U. stipitata*, *U. guatterioides*, *Duguetia flagelaris* and *Xylopia benthamii*). By means of gas chromatography coupled to mass spectrometry, in vitro cytotoxic and anti-lipase assays, principal component analysis and molecular docking, it was possible to establish the main compounds that may be responsible for the cytotoxic effect of *O. amazonicum* and *B. pleiosperma*. Moreover, the anti-lipase potential of *D. flagerallaris* was also established, as well as its composition related to the activity. Thus, by the employed strategy, *allo*-aromadendrene, cryptomerione, δ-cadinene and β-bisabolene were suggested as plausible cytotoxic agents against cancer cell lines, and dehydroaromadendrene, spathulenol and elemol, against lipase. The present study provides significant information on the chemical profile and bioactivity studies of Amazon Annonaceae aromatic plants.

**Keywords:** Annonaceae; essential oils; *Bocageopsis*; *Onychopetalum*; lipase; cancer

## 1. Introduction

The Amazon Region comprises an area of seven million square kilometers, covering the territory of several South American countries. This area hosts the Amazon rainforest, the richest ecosystem of the world, in which, approximately 16,000 plant species are catalogued, including a substantial proportion that are exclusive to this biome [1]. Among the Amazon Rainforest flora, several species are considered appropriate renewables sources for the commercial production of essential oils and flavors [2]. The most iconic example is pau-rosa tree (*Aniba rosaeodora* Duke) oil, a valuable raw material in the cosmetics industry [3]. Moreover, Amazon essential oils are reported to possess several biological activities, many of them with pharmacological and biotechnological applications [2,4,5].

Among the Amazonian plant producers of essential oils, those belonging to the family Annonaceae represent potential sources of biologically active compounds [6].

The chemistry of Annonaceae essential oils is complex, with the predominance of terpenes as main constituents, especially mono and sesquiterpenes [7]. Among important biological activities, Annonaceae essential oils present remarkable anticancer properties. As an example, regarding the essential oil of *Guatteria friesiana* (W.A. Rodrigues), Erkens and Maas showed in vitro and in vivo antitumor activities, which have lately been attributed to the presence of eudesmol isomers [8]. These sesquiterpenes were able to induce caspase-mediated apoptosis in human hepatocellular carcinoma cells (HepG2). Additionally, high levels of (*E*)-caryophyllene in *Xylopia frutescens* Aubl. were correlated to the panel of activities recorded [9]. Moreover, oils with high content of *β*-bisabolene from *Duguetia gardneriana* Mart. inhibited mouse melanoma cells (B16-F10) [10].

Despite the fact that research evaluating the cytotoxicity of Annonaceae essential oils has increased over the last year, nothing is known about their anti-diabetic potential. To date, only a previous record on a terpene-rich fruit fraction of an African *X. aethiopica* (Dunal) A. Rich. displayed in vivo anti-diabetic effects in a type 2 diabetes model of rats [11]. Regarding essential oils, significant *α*-glucosidase inhibition was observed for *Hertia cheirifolia* L. (> 60% of *β*-pinene) [12] and lipase inhibition for *Lippia turbinata* Griseb. (> 75% of limonene) [13], thereby indicating the pharmacological and biotechnological capabilities of essential oils. More recently, computational strategies, such as molecular docking, have been successfully employed to help understand how target enzymes related to cancer and diabetes interact with potential inhibitors, thus proving to be a powerful, rational and low-cost tool for the screening of bioactive natural products and synthetic compounds [14–16].

Therefore, the aim of this work was to evaluate the in vitro cytotoxic activities against a panel of human cancer cell lines, the anti-diabetic potentials in enzyme inhibition and anti-glycant assays of seven essential oils previously obtained from Amazonian Annonaceae species belonging to *Unonopsis*, *Bocageopsis* and *Onychopetalum* genera [17–19], along with essential oils from *Duguetia flagelaris* and *Xylopia benthamii*, which were analyzed by gas chromatography–mass spectrometry. Moreover, principal component analysis (PCA) and molecular docking were applied to study the main constituents of the active oils.

## 2. Materials and Methods

### 2.1. Chemicals

The reagents 2,2-diphenyl-1-picrylhydrazyl (DPPH), 2,2'-azino-bis(3-ethylbenzothiazoline-6-sulphonic acid) (ABTS) (≥98%), porcine pancreas lipase (Type II, 100–400 units/mg protein), porcine pancreas *α*-amylase (Type VI-B, from porcine pancreas, 100 units/mg protein), *α*-glucosidase from *Saccharomyces cerevisiae* (Type I; 10 U/mg protein), Folin and Ciocalteu's reagent, anhydrous $Na_2SO_4$, $NaH_2PO_4$, $Na_2HPO_4$, bismuth(III) nitrate pentahydrate, bovine serum albumin (BSA), sodium azide, methylglyoxal and fructose were purchased from Sigma-Aldrich (St. Louis, MO, USA). The linear $C_7$–$C_{30}$ alkanes mix was also purchased from Sigma-Aldrich (St. Louis, MO, USA). HPLC grade dichloromethane was from Tedia (Fairfield, OH, USA) and ultrapure water (18.2 MΩ) was obtained by purification with a Milli-Q gradient system (Millipore, Milford, MA, USA).

## 2.2. Plant Collection

The barks of *Duguetia flagelaris* (DFB) and *Xylopia benthamii* (XBB) were collected in March 2019 at the botanical garden of Manaus (59°56′24.3″ W 3°00′16.6″ S), Amazonas state, Brazil. Both specimens were previously listed in floristic projects. The herbarium deposit numbers of the collections are listed in the Supplemental Materials. The access to genetic heritage was registered at Sistema Nacional de Gestão do Patrimônio Genético e do Conhecimento Tradicional Associado (SisGen) under the code A8B52C3.

## 2.3. Essential Oil Extraction

After collection, the fresh plant barks were cleaned, manually cut into small pieces and dried at 40 °C in an air circulating stove for 24 h. Then, the plant materials were crushed in a metal blender and directly extracted by hydrodistillation with a Clevenger-type apparatus (DiogoLab, Poá, SP, Brazil). The amounts of starting material were standardized to 1 kg of powdered plant per 4 L of ultrapure water to ensure comparability. Furthermore, the obtained essentials oils were extracted three times with dichloromethane, dried over anhydrous $Na_2SO_4$ and filtered through a nylon membrane (pore size 0.22 μm, Whatman, Maidstone, UK). The essential oils were stored in a freezer (−15 °C) prior to analysis.

## 2.4. Cell Viability Assay

HepG2 (human hepatocellular carcinoma), HL-60 (human promyelocytic leukemia), K562 (human erythroleukemic) and PBMC (human peripheral blood mononuclear) cell lines were purchased from the American Type Culture Collection (ATCC, Manassas, VA, USA). All cell lines were cultured following the instructions in the ATCC animal cell culture guide and were tested for mycoplasma using a mycoplasma stain kit (Sigma-Aldrich, St. Louis, MO, USA) to validate the use of mycoplasma-free cells. Before each experiment, the number of viable cells was quantified by the trypan blue exclusion (TBE) assay. In summary, 10 μL of trypan blue (0.4%) was added to 90 μL of cell suspension and cells were counted using a Neubauer chamber.

To assess the cytotoxicity of essential oils, cell viability was measured by the Alamar blue assay, as previously described [5]. Briefly, cells were seeded in 96-well plates. The essential oils (ranging from 0.4 to 50 μg/mL), dissolved in dimethyl sulfoxide (DMSO, Vetec Química Fina Ltd.a, Duque de Caxias, RJ, Brazil), were added to each well and incubated for 72 h. Doxorubicin (ranging from 0.04 to 5 μg/mL) (purity ≥95%, doxorubicin hydrochloride, Laboratory IMA S.A.I.C., Buenos Aires, Argentina) and 5-fluorouracil (ranging from 0.2 to 25 μg/mL) (5-FU, purity > 99%; Sigma-Aldrich, St. Louis, MO, USA) were used as positive controls. At the end of the treatment, 20 μL of a stock solution (0.312 mg/mL) of resazurin (Sigma-Aldrich, St. Louis, MO, USA) was added to each well. Absorbances at 570 nm and 600 nm were measured using a SpectraMax 190 Microplate Reader (Molecular Devices, Sunnyvale, CA, USA). The values of half-maximum inhibitory concentration ($IC_{50}$) and their respective 95% confidence intervals were obtained by non-linear regression.

## 2.5. Lipase Inhibition Assay

Lipase inhibition activity was determined as previously described [20], with modifications. Initially, the samples were dissolved in DMSO at 10 μg/mL, then diluted until 1 μg/mL. Pancreas porcine lipase Type II (Sigma-Aldrich, St. Louis, MO, USA) was diluted in TRIZMA® Base buffer (Sigma-Aldrich, St. Louis, MO, USA) at 75 mM, pH 8.5 and *p*-nitrophenyl palmitate was used as a substrate (Sigma-Aldrich, St. Louis, MO, USA), which was diluted in acetonitrile and ethanol (1:4, *v/v*). The standard orlistat (Sigma-Aldrich, St. Louis, MO, USA) was used as the positive control. Then, 30 μL of test samples of the standard and/or DMSO were placed with 250 μL of lipase solution (0.8 μg/mL) in each well (in triplicate). The mixture was kept in the dark for 5 min at 37 °C; then, 20 μL of *p*-

nitrophenyl palmitate (4 μg/mL) were added. Readings were taken every 10 min or until the control reading reached an absorbance of 1000 ± 0.1. After determining the percentage of inhibition, samples with inhibition greater than 50% were diluted in serial concentrations (100, 80, 60, 40, 20, 10 and 5 μg/mL) to determine the $IC_{50}$. The analyses were carried on a microplate reader, model DTX 800 Multimode Detector (Beckman Coulter, Lane Cove, NSW, Australia), at 450 nm.

### 2.6. Chromatographic Analysis

The essential oils from barks of *Duguetia flagelaris* (DFB) and *Xylopia benthamii* (XBB) were analyzed by gas chromatography coupled to mass spectrometry (GC-MS) using a GCMS/QP2010 Plus apparatus (Shimadzu, Kyoto, Japan) equipped with a capillary column Rtx-5 MS (30 m × 0.25 mm × 0.25, Restek, Bellefonte, PA, USA). Helium at a flow of 1 mL/min was the carrier, and injections of 1 μL were performed with stock solutions at 1.0 mg/mL in dichloromethane with a split ratio of 1:50. The column temperature program was 60 to 285 °C, with gradual increases of 3 °C/min. The temperatures of the injector and the ion source were 215 °C and 260 °C, respectively. Identifications were based on comparisons of the obtained spectra with those stored in the Wiley 8th edition library (similarities <85% were discontinued) and by comparison of retention index (RI) with literature data [21]. RI was calculated according to Van Den Dool and Kratz equation [18,19] through the co-injection of a homologous series of linear *n*-alkane ($C_7$-$C_{30}$). A semi-quantitative analysis was performed to obtain the relative amount of each component of the essential oils. For this, gas chromatography with flame ionization detection (GC-FID) was applied. A system comprising a piece of GC2010 equipment (Shimadzu) equipped with an Rtx-5 capillary column was used, with the same conditions as the GC–MS analysis. Relative amounts (%) were calculated in relation to the total area of the chromatogram of independent triplicates.

### 2.7. Statistical Analysis

All chemical analyses and in vitro biological assays were run in triplicate and the results were expressed as mean and range of variation. The essential oils results were investigated with multivariate analysis through principal component analysis (PCA) with the software R version 4.1.1. For the construction of the PCA, only the active substances (with the percentage of representativeness) in each of the oils remained for the analysis.

### 2.8. Molecular Docking Analysis

Initially, the three-dimensional (3D) structures of the main compounds of the active essential oils were downloaded from PubChem database (https://pubchem.ncbi.nlm.nih.gov/ (accessed on 11 January 2022)) in spatial data file (SDF) format. The ligands were prepared according to previously reported methodologies [22]. Briefly, all the structures were subjected to geometry optimization by the semi-empirical method PM7 using MOPAC2016 software being the results saved in protein data bank (PDB) format. In the AutoDock Tools, Gasteiger charges were added for each compound and nonpolar hydrogens were merged, being the results saved in protein data bank, partial charge (Q) and Atom Type (T) (PDBQT) format [23,24].

The 3D crystal structures of tyrosine kinases of the epidermal growth factor receptor (EGFR-TK) (PDB ID: 1M17), FMS-like tyrosine kinase-3 (FLT3) (PDB ID: 4RT7), and porcine pancreatic lipase (PPL) (PDB ID: 1ETH) were retrieved from Research Collaboratory for Structural Bioinformatics Protein Data Bank (RCSB PDB) (http://www.rcsb.org accessed on 11 January 2022) in PDB format. In the AutoDock Tools, water molecules and bound ligands were removed, polar hydrogens and Kollman charges were added, and the non-polar hydrogens were merged. Finally, the results were saved as PDBQT format.

The ligand–enzyme interactions, as well as the binding affinity, were predicted using Autodock Vina [25]. A grid box of 24 × 24 × 24 Å in x, y, z directions was created with a

spacing of 1.00 Å, and centered at x = 21.683, y = 1.726, z = 54.278 for 1M17; a grid box of 30 × 30 × 30 Å in x, y, z directions was created with a spacing of 1.00 Å, and centered at x = −40.526, y = 11.236, z = −14.574 for 4RT7; and a grid box of 30 × 30 × 30 Å in x, y, z directions was created with a spacing of 1.00 Å, centered at x = 56.901, y = 47.835, z = 122.106 for 1ETH. The visual ligand–enzyme interactions were displayed using Discovery Studio Visualizer. Due to the lack of models with ligands for PPL in the RCSB PDB, only the FLT3 and EGFR-TK was tested for redocking.

## 3. Results and Discussion

The analysis of the essential oils from barks of *Duguetia flagelaris* (DFB) and *Xylopia benthamii* (XBB) showed that both samples were exclusively constituted by terpenoids (Table 1). The class and oxidation degree distributions displayed the predominance of sesquiterpenes, monoterpenes and diterpenes in minor amounts. This preliminary observance indicates that sesquiterpenes, both hydrocarbons and oxygenated versions, may be involved in any biological activities of the obtained essential oils. Monoterpenes were observed as traces in both samples, while diterpenes were detected solely in the latter. For the essential oil of *D. flagelaris* (DFB), spathulenol (11.9%), elemol (6,5%) and dehydroaromadendrene (6.1%) were observed as the main constituents. Despite being previously described as the main component in the stem-barks essential oil of *D. flagellaris* (> 58%), the presence of spathulenol was nearly three-fold less than previously stated [26]. On the other hand, for the essential oil of *X. benthamii* (XBB), (*E*)-caryophyllene was the main component (46.9%), followed by bicyclogermacrene (12.5%). These last findings are new, since only the flowers of this species have been the subject of essential oil analysis, for which methylbenzoate and 2-phenylethyl alcohol benzenoids were the main previously identified components [27].

**Table 1.** Essential oil composition of previously reported *Unonopsis*, *Bocageopsis* and *Onychopetalum* species, along with *Duguetia flagelaris* and *Xylopia benthamii*. In this table: UGL = leafs of *Unonopsis guatterioides*; USL = leafs of *U. stipitata*; UFL = leafs of *U. floribunda*; URL = leafs of *U. rufescens*; UDL = *U. duckei*; BPL = leafs of *Bocageopsis pleiosperma*; BPB = barks of *B. pleiosperma*; BPS = stems of *B. pleiosperma*; OAL = leafs of *Onychopetalum amazonicum*; OAB = barks of *O. amazonicum*; OAS = stems of *O. amazonicum*; XBB = barks of *Xylopia benthamii*; DFB = barks of *Duguetia flagellaris*.

| Scheme | UGL | USL | UFL | URL | UDL | BPL | BPB | BPS | OAL | OAB | OAS | XBB | DFB |
|---|---|---|---|---|---|---|---|---|---|---|---|---|---|
| Artemisia triene | - | - | - | - | - | - | - | - | - | - | - | - | 0.3 |
| α-Pinene | - | - | - | - | - | - | - | - | - | - | - | 3.3 | - |
| β-Pinene | - | - | - | - | - | - | - | - | - | - | - | 2.2 | - |
| α-Phellandrene | - | - | - | - | - | - | - | - | - | - | - | 14.1 | - |
| Limonene | - | - | - | - | - | - | - | - | - | - | - | 1.4 | - |
| Linalool | - | - | - | - | 0.3 | - | - | - | - | - | - | - | - |
| Verbenol | - | - | - | - | - | - | - | - | - | - | - | - | 1.7 |
| Citronellol | - | - | - | - | - | - | - | - | - | - | - | - | - |
| δ-Elemene | 2.51 | 1 | 0.27 | - | 0.48 | - | - | - | 0.7 | - | - | - | - |
| α-Cubebene | 1.19 | - | 0.83 | 0.24 | 0.34 | - | 1.62 | - | 0.7 | 1.3 | - | - | - |
| Citronellyl acetate | - | - | - | - | - | - | - | - | - | - | - | - | - |
| Cyclosativene | 0.71 | 0.11 | 0.27 | - | 0.4 | - | - | - | - | - | - | - | - |
| α-Ylangene | 0.34 | - | 0.17 | 0.31 | | - | - | - | - | - | - | - | - |
| Isoledene | - | - | - | - | - | - | - | - | - | - | - | - | 1.8 |
| α-Copaene | 11.26 | 0.4 | 6.26 | 3.71 | 1.99 | 1.16 | 3.28 | - | 8.4 | 3.4 | 0.2 | - | - |
| β-Bourbonene | 0.9 | - | 1.24 | 1.44 | 0.46 | - | - | - | 0.6 | - | - | - | - |
| β-Cubebene | 0.84 | - | 0.84 | 0.26 | 0.49 | - | 1.10 | - | - | - | - | - | - |
| β-Elemene | 2.03 | 1.31 | 0.89 | 0.42 | 2.65 | - | 0.88 | 0.22 | 2.7 | 2.9 | 4.2 | - | 1.6 |
| 7-*epi*-Sesquithujene | - | - | - | - | - | 0.56 | - | - | - | - | - | - | - |

| Compound | 1 | 2 | 3 | 4 | 5 | 6 | 7 | 8 | 9 | 10 | 11 | 12 | 13 |
|---|---|---|---|---|---|---|---|---|---|---|---|---|---|
| β-Longipinene | - | - | - | - | - | - | 0.47 | - | - | - | - | - | - |
| α-Gurjunene | - | - | 0.19 | - | 1.26 | - | - | - | 3.6 | 14.9 | 10.6 | - | 4.1 |
| (E)-Caryophyllene | 3.91 | 18.76 | 4.06 | 3.97 | 1.22 | - | 3.61 | - | 17.0 | 3.8 | - | 46.9 | - |
| β-Gurjunene | 0.76 | - | 0.15 | 0.32 | 0.8 | - | - | - | - | - | - | - | - |
| (E)-α-Bergamotene | - | - | - | - | - | 6.9 | 1.54 | 2.49 | - | - | - | - | 0.3 |
| γ-Elemene | 0.44 | - | 0.12 | - | - | - | - | - | - | - | - | - | - |
| α-Guaiene | - | - | 2.48 | 2.58 | - | - | - | - | - | - | - | - | - |
| Aromadendrene | 1.13 | 2.38 | - | - | 0.44 | - | - | 0.63 | 1.6 | - | - | - | - |
| (Z)-β-Farnesene | - | - | - | - | - | 2.09 | - | - | - | - | - | - | - |
| Sinularene | 0.18 | 0.21 | - | - | - | - | 0.98 | - | - | - | - | - | - |
| α-Neoclevene | - | - | - | - | - | - | - | - | - | - | - | - | 2.6 |
| α-Humulene | 1.74 | 5.18 | 0.94 | 0.84 | 0.96 | - | - | - | 3.1 | 0.9 | - | 4.5 | - |
| (E)-β-Farnesene | - | - | - | - | - | 6.05 | 1.24 | 1.04 | - | - | - | - | - |
| allo-Aromadendrene | 3.55 | - | - | - | 0.4 | - | 3.66 | - | - | 21.2 | 2.4 | - | - |
| Dehydroaromadendrene | - | - | - | - | - | - | - | - | - | - | - | - | 6.1 |
| γ-Gurjunene | - | - | - | - | - | - | - | - | - | 1.3 | - | - | 0.7 |
| γ-Muurolene | 2.79 | 0.45 | 4.18 | 6.4 | 1.11 | - | 1.70 | - | 2.9 | - | - | - | - |
| ar-Curcumene | - | - | - | - | - | - | - | - | - | - | - | - | - |
| Germacrene D | 1.62 | 4.56 | 1.54 | 2.59 | 1.24 | - | - | - | 1.7 | - | - | - | 1.4 |
| β-Selinene | 0.31 | 0.3 | 0.8 | 0.5 | 0.54 | 0.29 | 6.46 | 0.39 | 0.7 | 0.7 | 1.4 | - | - |
| δ-Selinene | - | 0.22 | 0.53 | - | - | - | - | - | - | - | - | - | - |
| (E)-Muurola-4(14)-5-diene | - | - | - | - | - | - | 0.35 | - | - | - | - | - | - |
| Valencene | 1.94 | - | 2.1 | - | 0.4 | - | - | - | - | - | - | - | - |
| α-Selinene | - | - | - | - | - | - | 5.2 | - | - | - | - | - | - |
| Bicyclogermacrene | - | 20 | 1.55 | 3.7 | - | - | - | - | 5.4 | - | - | 12.5 | 0.6 |
| α-Muurolene | 1.26 | 0.23 | 1.93 | 2.38 | 0.57 | 0.25 | 0.96 | - | 1.2 | 0.7 | - | - | 0.9 |
| β-Bisabolene | - | - | - | - | - | 55.71 | 38.53 | 34.37 | - | - | - | - | - |
| (E,E)-α-Farnesene | - | - | - | - | - | - | - | - | - | - | 0.4 | - | - |
| α-Bulnesene | - | 0.4 | 1.69 | 1.24 | - | - | - | - | - | - | - | - | 1.3 |
| γ-Cadinene | 1.91 | 0.26 | 4.96 | 4.26 | 0.92 | | 1.60 | - | - | 1.9 | 1.8 | 1.4 | 0.4 |
| (E)-Calamene | - | - | - | - | - | - | - | - | - | - | 1.9 | - | - |
| δ-Cadinene | 2.24 | 0.7 | 5.37 | 4.02 | 0.83 | 1.42 | 7.55 | - | 5.9 | 3.9 | 2.4 | - | - |
| (E)-γ-Bisabolene | - | - | - | - | - | - | - | 0.62 | - | - | - | - | - |
| (E)-Cadina-1,4-diene | - | - | 0.43 | - | - | - | - | - | - | - | - | - | - |
| α-Cadinene | 0.25 | - | 0.39 | 0.55 | - | - | - | - | - | - | - | - | - |
| α-Calacorene | - | - | 2.16 | 2.56 | - | 1.00 | 0.52 | - | 2.7 | - | 1.3 | - | 3.3 |
| Elemol | 1.6 | 1.29 | 1.3 | 0.68 | 7.86 | - | - | - | - | - | 2.5 | - | 6.5 |
| β-Vetivenene | 0.63 | 0.53 | 1.14 | 1.09 | 1.29 | - | - | - | - | - | - | - | - |
| Germacrene B | 1.84 | 0.56 | 0.52 | 0.29 | - | - | 0.32 | - | 1.0 | - | - | - | 1.6 |
| (E)-Nerolidol | 0.52 | - | - | - | 0.63 | - | - | - | - | - | - | - | - |
| β-Calacorene | - | - | - | - | - | - | - | - | 0.8 | - | - | - | - |
| 3-(Z)-Hexenyl benzoate | - | - | - | - | 0.53 | - | - | - | - | - | - | - | - |
| 1.5-Epoxysalvial-4(14)-ene | 0.46 | 0.88 | 0.3 | 0.26 | 3.65 | - | - | - | - | - | - | - | - |
| Dendrolasin | - | - | - | - | - | - | - | 0.26 | - | - | - | - | - |
| (E)-Sesquisabinene hydrate | - | - | - | - | - | - | - | - | - | - | 0.5 | - | - |
| Spathulenol | 4.8 | 20.5 | 15.7 | 17.13 | 19.1 | - | - | - | 10.4 | - | - | - | 11.9 |
| Caryophyllene oxide | 4.8 | 8.41 | 9.77 | 15.9 | 9.7 | 2.8 | 3.00 | - | 11.9 | 3.7 | 4.9 | 0.8 | 2.5 |
| ar-Turmerol | - | - | - | - | - | - | - | 0.72 | - | - | - | - | - |
| Gleenol | - | - | - | - | - | - | 0.34 | - | - | - | - | - | - |
| β-Copaen-4-α-ol | 0.32 | 0.27 | 0.75 | 0.56 | - | - | - | - | - | - | - | - | - |
| Viiridiflorol | 1.61 | 1.6 | 0.99 | 0.84 | - | - | - | - | 0.7 | 0.9 | 1.4 | 1.5 | 2.0 |
| Salvial-4(14)-en-1-one | - | 0.47 | - | - | 0.25 | - | - | 0.31 | - | - | - | - | - |
| Widdrol | - | - | 1.14 | 0.63 | - | - | - | - | - | - | - | - | 4.1 |
| Guaiol | 5.14 | 0.85 | 1.43 | 1.57 | 6.41 | - | - | - | 1.1 | - | - | - | - |
| Rosifoliol | - | 0.8 | - | - | - | - | - | - | - | - | - | - | - |
| Ledol | - | - | - | - | 1.83 | - | - | - | - | - | - | 4.6 | 3.6 |
| Sesquithuriferol | - | - | - | - | - | - | - | - | - | - | 0.7 | - | - |

| Compound | | | | | | | | | | | | | |
| --- | --- | --- | --- | --- | --- | --- | --- | --- | --- | --- | --- | --- | --- |
| Humulene epoxide II | 0.95 | 0.98 | 2.26 | 2.89 | 3.21 | - | 0.89 | - | 1.1 | - | 3.4 | - | - |
| β-atlanol | - | - | - | - | - | - | - | 4.09 | - | - | - | - | - |
| 1,10-di-*epi*-Cubenol | - | - | - | - | - | - | - | - | - | - | 3.2 | - | - |
| Isolongifolan-7-α-ol | - | - | - | - | - | - | - | - | - | - | 2.4 | - | - |
| 1-*epi*-Cubenol | - | - | - | - | - | 0.41 | 3.55 | - | - | 3.2 | 2.8 | - | - |
| Isospathulenol | 5.51 | 1.6 | 3.14 | 2.21 | 2.57 | - | - | - | - | - | - | - | - |
| α-*epi*-Cadinol | - | - | - | - | - | - | - | - | - | 24.1 | 14.0 | 2.9 | - |
| Epoxy-*allo*-aromadendrene | - | - | - | - | - | 0.60 | - | - | - | - | - | - | - |
| Cubenol | - | - | - | - | - | - | 3.2 | - | 1.4 | 0.8 | - | - | - |
| α-Muurolol | 2.45 | - | 1.35 | 0.78 | 1.37 | 0.22 | 0.97 | - | - | - | - | - | - |
| Torreyol | - | - | - | - | - | - | - | - | - | - | - | - | 1.8 |
| β-Eudesmol | 1.13 | - | 1.1 | 1.16 | 1.49 | - | - | - | - | - | - | - | 2.3 |
| α-Cadinol | 3.64 | 0.59 | 1.62 | 2.09 | 3.05 | 0.47 | 0.37 | - | 0.5 | 1.3 | 0.7 | - | - |
| α-Bisabolol oxide B | - | - | - | - | - | - | - | 1.43 | - | - | - | - | - |
| 14-Hydroxy-(*Z*)-caryophyllene | - | - | - | - | - | 0.36 | - | - | - | - | - | - | - |
| β-Atlantone | - | - | - | - | - | - | - | 0.74 | - | - | - | - | - |
| *E*-calamen-10-ol | - | - | - | - | - | 0.23 | - | - | - | - | - | - | - |
| Bulnesol | 2.65 | - | 0.7 | 0.8 | 2.4 | - | - | - | - | - | - | - | - |
| β-Bisabolol | - | - | - | - | - | 0.47 | - | 0.94 | - | - | - | - | - |
| Cadalene | - | - | 0.48 | 0.62 | 0.52 | 0.22 | - | - | - | - | 0.9 | - | - |
| Mustakone | - | - | - | - | - | - | - | - | - | - | 1.1 | - | - |
| α-Bisabolol | - | - | - | - | - | 1.45 | 2.21 | 1.9 | - | - | - | - | - |
| Eudesma-4(15),7-dien-1-β-ol | - | - | - | - | - | - | - | - | - | - | 1.8 | - | - |
| Cyperotundone | - | - | - | - | - | - | - | - | - | 1.3 | 8.1 | - | - |
| (2*Z*.6*Z*)-Farnesol | - | - | - | - | - | - | - | 7.2 | - | - | - | - | - |
| β-Sinensal | - | - | - | - | - | - | - | 0.35 | - | - | - | - | - |
| (2*E*.6*Z*)-Farnesol | - | - | - | - | - | 0.98 | - | 3.54 | - | - | - | - | - |
| Cryptomerione | - | - | - | - | - | 2.58 | 1.03 | 9.6 | - | - | - | - | - |
| (2*E*.6*E*)-Farnesol | - | - | - | - | - | - | - | 0.5 | - | - | - | - | - |
| β-Bisabolenal | - | - | - | - | - | - | - | 0.62 | - | - | - | - | - |
| β-Bisabolenol | - | - | - | - | - | 0.64 | - | 0.2 | - | - | - | - | - |
| (2*E*.6*E*)-Farnesyl acetate | - | - | - | - | 0.24 | - | - | - | - | - | - | - | - |
| (5*E*.9*E*)-Farnesyl acetone | 1.52 | - | - | - | - | - | - | - | - | - | - | - | - |
| Phytol | - | - | - | - | - | - | - | - | - | - | - | - | 3.4 |
| Cembrene A | - | - | - | - | - | - | - | - | - | - | - | - | 0.4 |

Regarding the other essential oil samples subjected to in vitro cytotoxic and anti-lipase assays, spathulenol, caryophyllene oxide and (*E*)-caryophyllene (Table 1) were previously reported as the main compounds in *Unonopsis* spp. (UGL, USL, UFL, URL and UDL) and *O. amazonicum* (OAL) [17,19]. On the other hand, the essential oils of *Bocageopsis pleiosperma* were pointed as promising sources of β-bisabolene [18]. Moreover, the δ-cadinene was present in most of the *Unonopsis*, *Onychopetalum* and *Bocageopsis* essential oils (except for BPS), being more representative in the BPB essential oil (7.55%). This substance was described as the majority in the essential oils of *Kadsura longipedunculata* (Schisandraceae), with 21.79% representativeness, and the oil of this plant showed antibacterial (against Gram-positive bacteria), trypanocidal (IC$_{50}$ = 50.52 μg/mL), and cytotoxic (cell lines MIA PaCa-2, HepG-2, and SW-480) activity [28]. α-*epi*-cadinol, another important bioactive compound, was reported in OAB and OAS essential oils (24.1% and 14%) [19]. A previous study showed that the essential oils from the leaves of *Jatropha curcas* (Euphorbiaceae), which contain a significant amount (7.38%) of this compound, has antioxidant and antimicrobial activities [29]. (2*Z.6Z*)-Farnesol, previously reported in BPS oils (7.20%) was previously associated with cytotoxic and antimicrobial activities [30]. Furthermore, cryptomerione was also previously reported as representative in the essential oils of *Bocageopsis* spp. (BPL, BPB and BPS). The biological activity of this substance has not yet been reported in the literature.

### 3.1. Cytotoxic Evaluation of Essential Oils

Among the evaluated essential oils, five displayed cytotoxic activities (Table 2). Despite different cell lines being employed in this study, the results were more significant with HL-60. Sample OAB was the sole sample that was active against two cell lines, with moderate cytotoxicity against HL-60 and K652 ($IC_{50}$ = 10.32 µg/mL and 17.70 µg/mL, respectively). Essential oils BPB and BPL displayed activities in the same range against HL-60, with $IC_{50}$ of 15.22 and 11.74 µg/mL, respectively. Considerable cytotoxicity was recorded for BPS ($IC_{50}$ = 8.70 µg/mL) and OAS ($IC_{50}$ = 5.36 µg/mL), the latter being the most active sample. Moreover, all assayed samples displayed no cytotoxicity or low cytotoxicity (30–50 µg/mL range) to normal cells (PBMC) (Table 2).

**Table 2.** Cytotoxic activity of essential oil evaluated.

| Samples | Cell Lines $IC_{50}$ (in µg/mL) and 95% CI (in µg/mL) | | | |
|---|---|---|---|---|
| | HepG2 | HL-60 | K562 | PBMC |
| *U. duckei* (leaves) (UDL) | >50 | >50 | >50 | >50 |
| *U. floribunda* (leaves) (UFL) | >50 | >50 | >50 | >50 |
| *U. guatterioides* (leaves) (UGL) | >50 | >50 | >50 | >50 |
| *U. rufescens* (leaves) (URL) | >50 | >50 | >50 | >50 |
| *U. stipitata* (leaves) (USL) | >50 | >50 | >50 | >50 |
| *O. amazonicum* (leaves) (OAL) | >50 | >50 | >50 | >50 |
| *O. amazonicum* (barks) (OAB) | 40.37 34.38–47.39 | 10.32 7.13–14.93 | 17.70 13.01–24.08 | 46.02 38.70–54.73 |
| *O. amazonicum* (stems) (OAS) | 28.58 25.00–32.67 | 5.36 4.23–6.79 | 29.73 23.53–37.55 | 33.47 28.20–39.71 |
| *B. pleiosperma* (barks) (BPB) | 35.93 30.96–41.69 | 15.22 10.63–21.80 | 20.33 12.12–34.12 | >50 |
| *B. pleiosperma* (leaves) (BPL) | >50 | 11.74 9.65–14.29 | 37.92 31.72–45.33 | 43.65 37.02–51.48 |
| *B. pleiosperma* (stems) (BPS) | 43.41 35.08–53.72 | 8.70 5.19–14.57 | 28.93 20.62–40.59 | 33.32 28.34–39.18 |
| *D. flagellaris* (barks) (DFB) | >50 | >50 | - | - |
| *X. benthamii* (barks) (XBB) | >50 | >50 | - | - |
| Doxorubicin | 0.46 0.33–0.62 | 0.06 0.05–0.08 | 0.33 0.19–0.59 | 0.67 0.49–0.91 |

Previous findings already indicate the antitumor potentials of essential oils from Annonaceae species [31]. Recently, the anti-leukemic potential of the essential oil from the leaves of *Guatteria megalophylla* (> 28% of spathulenol) was evaluated in both in vitro and in vivo experiments, showing promising results [31]. Moreover, a *β*-bisabolol-rich (> 80%) essential oil from *Duguetia gardneriana* displayed cytotoxic activities against a panel of cancer cell lines with $IC_{50}$ ranging from 10 to 20 µg/mL, including HL-60 (13.1 µg/mL) [10]. Considering the evaluated species, only *X. benthamii* had its cytotoxic capabilities evaluated previously [32]. Since all the essential oils exhibited sesquiterpenes as main constituents, the hypothesis of such compounds being the active ingredients was further investigated.

### 3.2. Lipase Enzyme Inhibition Assessment

The Annonaceae family is also seen as a relevant source of extracts with antidiabetic activity or as an inhibitor of diabetes-related enzymes [33]. Therefore, knowing that lipase inhibitors prevent the digestion and intestinal absorption of dietary fats, helping to fight obesity and diabetes, the assessment of oil samples regarding the inhibitory potentials against the lipase enzyme was considered.

Among the essential oil samples tested in our study, the only one that showed anti-lipase activity was *D. flagerallaris* (IC$_{50}$ = 93.6 ± 2.2 µg/mL), whose profile was comparable to the standard orlistate (IC$_{50}$ of 93.2 µg/mL) [34]. Through the analysis of the sample screening, we suggest that the inhibitory activity found for this sample, may be correlated to the presence of spathulenol, which, for *D. flagerallaris* oil, counts as the main component (11.9%, see Table 1). However, we observed similar concentrations of this substance in non-active oils, such as in *Unonopsis* spp. This is the first record that evaluates the potential of anti-lipase activity for *D. flagerallaris*. A previous study showed that the essential oil of three species of the *Stachys* genus (*S. inflata, S. lavandulifolia* and *S. byzantina*), which have many monoterpenes in their compositions, had a potential enzyme inhibitory response [35]. Another study showed that $\gamma$-terpinene, $\alpha$-terpinene and *p*-cimene are representative in oil from *Origanum vulgare,* and $\alpha$-limonene from *Lippia turbinata* [13]. Both oils in this study showed intense anti-lipase activity (IC$_{50}$ 5.09 and 7.26 µg/mL, respectively).

Although there are several studies for essential oils of Annonaceae, especially species of *Xylopia*, only a few were assayed with the aim of assessing their anti-diabetic potential. Among them, *X. aethiopica* showed a notable anti-diabetic profile [11]. Furthermore, the extract of *Duguetia furfuraceae* showed a considerable hypoglycemic effect [36] and was also able to inhibit the lipase enzyme.

Principal Component Analysis (PCA)

Through the ordering performed by the PCA, it was possible to observe that the first two dimensions explain more than half of the variation found in the samples (29.07% and 26.69%, respectively). The samples were grouped by similarity of chemical constituents, and it was possible to observe that BPB and BPL are chemically closer, as well as OAS and OAB, since they are positioned in similar regions in the graph. On the other hand, the BPS and DFB samples are located isolated in distinct regions of the graph, demonstrating that they are chemically more different than the others (Figure 1).

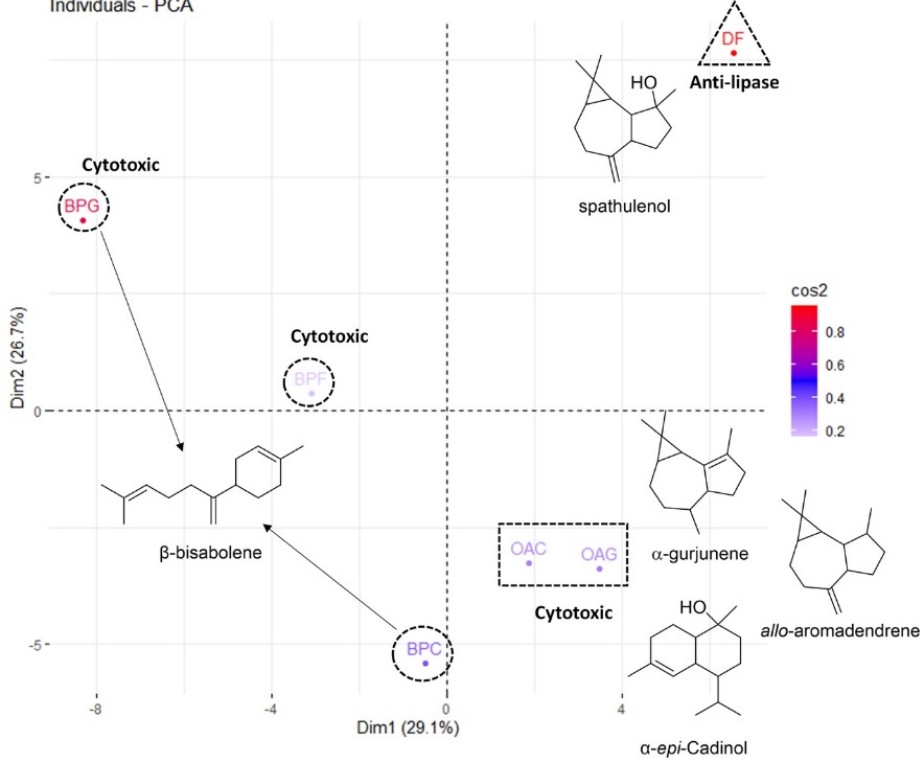

**Figure 1.** Ordering graph of the main components of the different samples of essential oils tested in this study. Colors in more intense red have a greater influence on ordering while more lavender colors are less influential.

The highest percentages of sample contribution were for BPS (50.38%) and DFB (31.19%) in the first dimension, and DFB (46.25%), BPB (23.10%) and BPS (13.01%) in the second dimension (Table 3).

**Table 3.** Percentage of contribution of each of the samples in the principal component analysis (PCA).

| Samples | Dimension 1 | Dimension 2 |
|---------|-------------|-------------|
| BPL | 6.90 | 0.10 |
| BPB | 0.17 | 23.10 |
| BPS | 50.38 | 13.01 |
| OAB | 2.53 | 8.44 |
| OAS | 8.80 | 9.07 |
| DFB | 31.19 | 46.25 |

We verified those compounds that presented themselves as major and unique in each of the oils, and we considered as significant the components that presented a percentage greater than 4% in relation to the total composition of each oil. As previously mentioned, *β*-bisabolene is a compound that is only present in samples from *Bocageopsis pleiosperma* (Table 1), showing the highest percentage of prevalence in leaves (55.71%), followed by bark (38.73%) and stems (34.37%). Additionally, due to the high predominance in the samples, we suggest *β*-bisabolene is responsible for the observed cytotoxicity against HL-60 (Table 2). *β*-bisabolene was previously described as possessing anticancer activity in in vivo and in vitro experiments, as well as antibacterial activity, being considered a promising substance in these lines of research [37,38].

Spathulenol was found only in *Dugetia flagellaris* essential oil, presenting itself as the major component (11.9%, Table 1). Despite spathulenol having records of biological activity, in this study, no cytotoxic activity was evidenced for the tested strains. The constituents found only in *Onychopetalum amazonicum* essential oils were *α*-gurjunene, *allo*-aromadendrene and *α*-epicadinol. Despite being present in both samples of *O. amazonicum*, these constituents showed differences in concentrations depending on the part of the plant used to obtain the essential oils. All these three constituents showed higher concentrations in the oil obtained from the barks (OAB), with *α*-gurjunene presenting 14.9%, *allo*-aromadendrene, 21.2% and *α*-epicadinol, 24.1%. For the oil from the stems (OAS), it was observed that *α*-gurjunene represented 10.6%, *allo*-aromadendrene, 2.6% and *α*-epicadinol, 14.0% (Table 1).

As higher cytotoxic activity was observed in OAS (Table 2), despite lower concentrations of the exclusive constituents in the oils, we suggest that the constituents *α*-gurjunene and *α*-epicadinol may be more related to the inhibitory activities, due to their high concentration in the active samples. Our results corroborate those found in the literature, where sesquiterpenes have been reported to have great potential for biological activity [9,39]. In addition to their cytotoxic activity, it is known that sesquiterpenes have antiparasitic [40] and antimalarial [39] activities.

### 3.3. Molecular Docking

Tyrosine kinases of the epidermal growth factor receptor (EGFR-TK), as well as FMS-like tyrosine kinase-3 (FLT3), are involved in cancer proliferation, the latter being particularly relevant in acute myeloid leukemia, and their inhibition represents a rational approach for the development of novel anticancer therapies [41,42]. On the other hand, porcine pancreatic lipase (PPL) inhibition has been described as a promising approach for the development of anti-obesity drugs [43]. Therefore, a preliminary docking study was conducted with the compounds suggested by PCA analysis as potential anticancer agents, as well as EGFR-TK, FLT3 and PPL.

Docking analysis applied to the EGFR-TK protein (Table 4) revealed that *allo*-aromadendrene (−7.4 kcal/mol) have close binding free energies to the cryptomerione (−7.1

kcal/mol) and δ-cadinene (−7.0 kcal/mol), while doxorubicin presented higher scoring function values (−9.8 kcal/mol).

**Table 4.** Docking results for the main constituents of the active essential oils.

| Compounds | Protein (PDB ID) | Binding Energy (kcal/mol) | Main Interactions |
|---|---|---|---|
| Doxorubicin [a] | | −9.8 | HB (Lys721, Thr766, Met769, Cys773, Arg817, Asn818, Thr830, Asp831), PSi (Phe699), PAl (Val702, Leu820) |
| *all*o-aromadendrene | | −7.4 | Al (Leu694, Val702, Ala719, Lys721, Leu820) |
| cryptomerione | | −7.1 | HB (Cys751), PAl (Ala719, Leu820), Al (Phe699, Val702, Ala719, Lys721, Met742, Leu764,) |
| δ-cadinene | Tyrosine kinase (1M17) | −7.0 | Al (Leu694, Val702, Ala719, Lys721, Leu768, Met769, Leu820) |
| (−)-β-bisabolene | | −6.8 | Al (Val702, Ala719, Lys721, Met742, Leu764, Leu768, Met769, Leu820) |
| (+)-β-bisabolene | | −6.8 | Al (Val702, Ala719, Lys721, Met742, Leu764, Leu768, Met769, Leu820) |
| *epi*-α-cadinol | | −6.8 | HB (Asp831), Al (Leu694, Val702, Ala719, Leu768, Leu820) |
| α-gurjunene | | −6.8 | Al (Val702, Ala719, Lys721, Cys773, Leu820) |
| doxorubicin [a] | | −8.1 | HB (Val808, Asp829), PAl (Ile836), PAm (Gly831) |
| (-)-β-bisabolene | | −9.0 | PSi (Phe691), PAl (Val624, Phe830), Al (Leu616, Val624, Ala642, Lys644, Val675, Phe691, Tyr693, Leu818, Cys828, Phe830) |
| (+)-β-bisabolene | FMS-like tyrosine kinase-3 (4RT7) | −8.9 | PSi (Phe691), PAl (Val624, Phe830), Al (Leu616, Val624, Ala642, Lys644, Val675, Phe691, Tyr693, Leu818, Cys828, Phe830) |
| cryptomerione | | −8.6 | PAl (Leu616, Ala642, Leu818), Al (Leu616, Val624, Ala642,Val675, Phe691, Tyr693, Leu818, Cys828) |
| δ-cadinene | | −6.9 | Al (Met664, Met665, Leu668, Val675, Leu802, Cys807, Cys828) |
| *all*o-aromadendrene | | −6.7 | PAl (Met664), Al (Met665, Leu668, Ile674, Val675, Phe691, Leu802, Ile827, Cys828) |
| *epi*-α-cadinol | | −6.2 | PAl (Met665), Al (Ile674, Val675, His809, Cys828) |
| α-gurjunene | | −6.2 | Al (Met664, Met665, Val675, His809, Ile827, Cys828) |
| orlistat [a] | Porcine pancreatic lipase (1ETH) | −7.2 | HB (Gly77, Phr78, Asp80, His152, Ser153), PSi (Tyr115), Al (Arg257, Ile210), AC (Asp80) |
| dehydroaromadendrene | | −8.7 | PAl (Phe78, Tyr115, Phe216, His264), Al (Ala179, Pro181) |
| spathulenol | | −7.7 | PAl (Phe78, Tyr115, Leu154), Al (Phe78, Ile79, Tyr115, Phe216, Val260, His264) |
| elemol | | −7.0 | PSi (Tyr115), PAl (Phe78, His152, Phe216, His264), Al (Ile79, Val260,  Ala261, Leu265) |

[a] Positive control; HB—hydrogen bond, PSi—π-sigma, PAl—π-alkyl, Al—alkyl, PAm—π-amide, AC—attractive charge.

Regarding to the observed interactions, hydrogen bonds were dominant to the doxorubicin, especially with carbonyl groups, highlighting the benzoquinone carbonyl interactions with Met769 and Cys773, and carbonyl interaction with Lys721. Hydroxyl interaction with Thr766 and amine interaction with Asn818 and Arg817 were also observed. These observations are in accordance with previously published data [44]. On the other hand, aromadendrene, cryptomerione and δ-cadinene presented, mainly, alkyl and π-alkyl interactions with some of these residues, many of which were through interactions with methyl groups or double bonds (Figure 2).

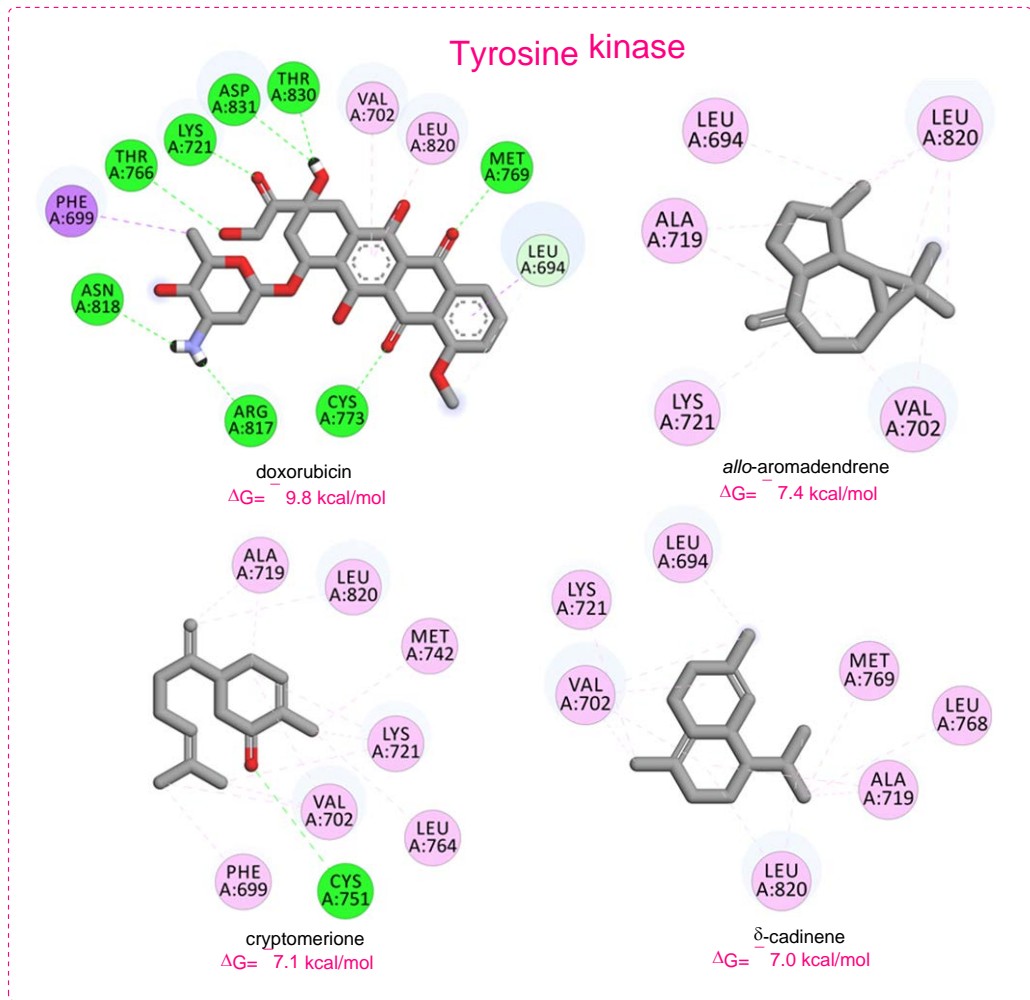

**Figure 2.** Main interactions observed for doxorubicin (positive control) and the top-scored inhibitors of tyrosine kinase (1M17) by docking analysis.

Docking analysis applied to the FLT3 protein (Table 4) revealed scoring function values close for (−)-$\beta$-bisabolene (−9.0 kcal/mol), (+)-$\beta$-bisabolene (−8.9 kcal/mol), and cryptomerione (−8.6 kcal/mol). Surprisingly, these binding free energies were best than the positive control doxorubicin (−8.1 kcal/mol). Doxorubicin presented hydrogen bonds through hydroxyl and amine groups with Asp829 and Val808, respectively, $\pi$-alkyl interaction between the methoxylated aromatic ring and Ile836, and $\pi$-amide interaction between the benzoquinone carbonyl and Gly831. Asp829, Phe830 and Gly831 are DFG-loop residues, and together with the hinge residue Val624, represent key residues at the FLT3 binding site [45]. For top-scored inhibitors, $\pi$-sigma, $\pi$-alkyl and alkyl interactions were observed between methyl groups or double bonds and these residues (Figure 3), suggesting these compounds exhibit favorable interactions to inhibit FLT3.

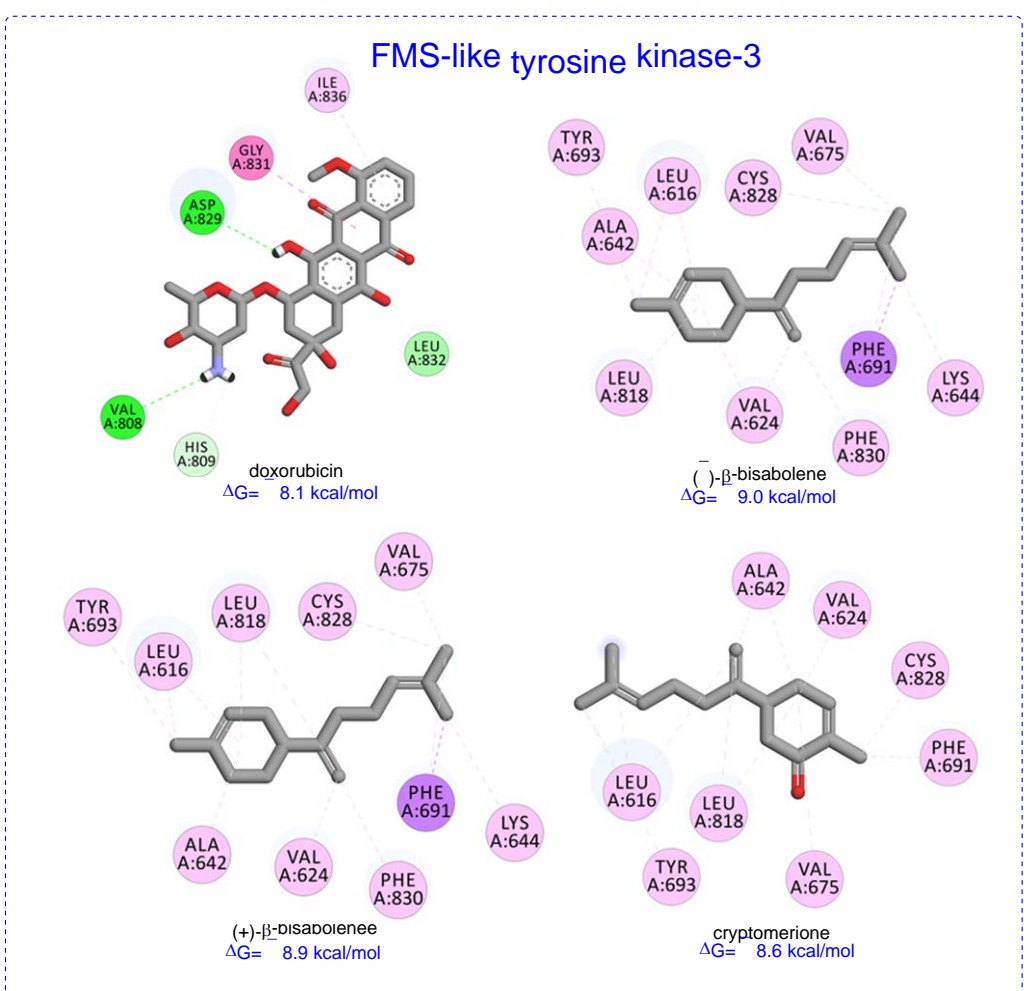

**Figure 3.** Main interactions observed for doxorubicin (positive control) and the top-scored inhibitors of FMS-like tyrosine kinase-3 (4RT7) by docking analysis.

Docking analysis applied to the PPL protein revealed that dehydroaromadendrene (−8.7 kcal/mol) and spathulenol (7.7 kcal/mol) have better binding free energies than the positive control orlistat (−7.2 kcal/mol). The observed interactions for orlistat were close to that previously described in the literature, which suggests a good approximation of the present model with those previously reported in the literature [46,47]. Orlistat presented hydrogen bonds through lactone carbonyl group with Ser153, hydroxyl group with Gly77 and His152, and amide group with Phe78. π-sigma interaction with Tyr115, alkyl interaction with Arg257 and Ile210, and an attractive charge interaction with Asp80 were also observed (Figure 4). The hydrogen bond with Ser153 is a key interaction since this residue along with Asp177 and His264 constitutes the catalytic triad of the PPL active site [48,49]. On the other hand, dehydroaromadendrene and spathulenol, as well as elemol, presented several π-alkyl and alkyl interactions, highlighting alkyl-based interaction with the key residue His264 from the catalytic triad (Figure 4). Overall, the molecular docking partially supports the results observed through biological assays.

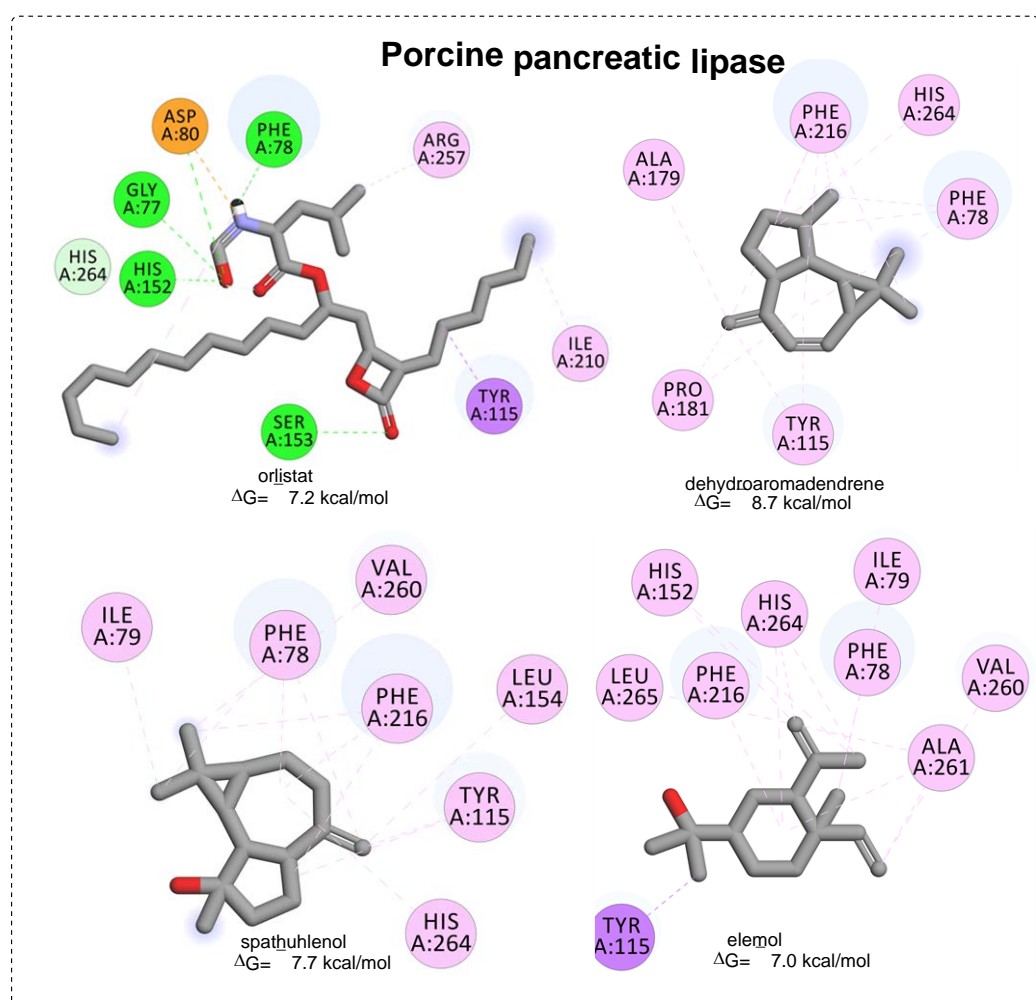

**Figure 4.** Main interactions observed for orlistat (positive control) and the top-scored inhibitors of porcine pancreatic lipase (1ETH) by docking analysis.

### 4. Conclusions

In conclusion, we have identified the volatile constituents of *D. flagelaris* and *X. benthamii* barks essential oil and evaluated the in vitro cytotoxic activities against HepG2, HL-60, K562 and PBMC cell lines, as well as anti-lipase activity. The chemical composition of both essential oils was similar to other Amazonian Annonaceae species previously investigated, being predominantly constituted by monoterpenes and sesquiterpenes. Regarding cytotoxic evaluation, our results clearly showed that *O. amazonicum* and *B. pleiosmerma* essential oils were more active against human promyelocytic leukemia cell line (HL-60), with IC$_{50}$ values ranging from 5.36 to 15.22 μg/mL. These cytotoxic properties were explained, in part, by the presence of *allo*-aromadendrene, cryptomerione, δ-cadinene and β-bisabolene. On the other hand, the anti-lipase potential of *D. flagerallaris* was established, and was related to the presence of dehydroaromadendrene, spathulenol and elemol. Additionally, molecular docking allowed us to understand how these compounds interact with key enzymes related to cancer and diabetes. However, further studies aimed at determining the anticancer and anti-lipase properties of these constituents are necessary to fully understand their bioactivities. Furthermore, our results indicate the importance of studies with Amazonian plants aiming to enhance the rational use of the immense biodiversity that is nowadays under permanent danger.

**Supplementary Materials:** The following supporting information can be downloaded at: https://www.mdpi.com/article/10.3390/chemistry4040081/s1, Figure S1: Main interactions observed between doxorubicin and tyrosine kinase (1M17) by docking analysis; Figure S2: Main interactions observed between *allo*-aromadendrene and tyrosine kinase (1M17) by docking analysis; Figure S3: Main interactions observed between cryptomerione and tyrosine kinase (1M17) by docking analysis; Figure S4: Main interactions observed between δ-cadinene and tyrosine kinase (1M17) by docking analysis; Figure S5: Main interactions observed between (-)-β-bisabolene and tyrosine kinase (1M17) by docking analysis; Figure S6: Main interactions observed between (+)-β-bisabolene and tyrosine kinase (1M17) by docking analysis; Figure S7: Main interactions observed between epi-α-cadinol and tyrosine kinase (1M17) by docking analysis; Figure S8: Main interactions observed between α-gurjunene and tyrosine kinase (1M17) by docking analysis; Figure S9: Main interactions observed between doxorubicin and FMS-like tyrosine kinase-3 (4RT7) by docking analysis; Figure S10: Main interactions observed between (-)-β-bisabolene and FMS-like tyrosine kinase-3 (4RT7) by docking analysis; Figure S11: Main interactions observed between (+)-β-bisabolene and FMS-like tyrosine kinase-3 (4RT7) by docking analysis; Figure S12: Main interactions observed between cryptomerione and FMS-like tyrosine kinase-3 (4RT7) by docking analysis; Figure S13: Main interactions observed between δ-cadinene and FMS-like tyrosine kinase-3 (4RT7) by docking analysis; Figure S14: Main interactions observed between *allo*-aromadendrene and FMS-like tyrosine kinase-3 (4RT7) by docking analysis; Figure S15: Main interactions observed between epi-α-cadinol and FMS-like tyrosine kinase-3 (4RT7) by docking analysis; Figure S16: Main interactions observed between α-gurjunene and FMS-like tyrosine kinase-3 (4RT7) by docking analysis; Figure S17: Main interactions observed between orlistat and porcine pancreatic lipase (1ETH) by docking analysis; Figure S18: Main interactions observed between dehydroaromadendrene and porcine pancreatic lipase (1ETH) by docking analysis; Figure S19: Main interactions observed between spathulenol and porcine pancreatic lipase (1ETH) by docking analysis; Figure S20: Main interactions observed between elemol and porcine pancreatic lipase (1ETH) by docking analysis; Figure S21: Exsiccate of *Unonopsis guatterioides* deposited in the herbarium of the Instituto Nacional de Pesquisas da Amazônia (INPA); Figure S22: Exsiccate of *Unonopsis stipitata* deposited in the herbarium of the Instituto Nacional de Pesquisas da Amazônia (INPA); Figure S23: Exsiccate of *Unonopsis floribunda* deposited in the herbarium of the Instituto Nacional de Pesquisas da Amazônia (INPA); Figure S24: Exsiccate of *Unonopsis rufescens* deposited in the herbarium of the Instituto Nacional de Pesquisas da Amazônia (INPA); Figure S25: Exsiccate of *Unonopsis duckei* deposited in the herbarium of the Instituto Nacional de Pesquisas da Amazônia (INPA); Figure S26: Exsiccate of *Bocageopsis pleiosperma* deposited in the herbarium of the Instituto Nacional de Pesquisas da Amazônia (INPA); Figure S27: Exsiccate of *Onychopetalum amazonicum* deposited in the herbarium of the Instituto Nacional de Pesquisas da Amazônia (INPA); Figure S28: Exsiccate of *Xylopia benthamii* deposited in the herbarium of the Instituto Nacional de Pesquisas da Amazônia (INPA); Figure S29: Exsiccate of *Duguetia flagellaris* deposited in the herbarium of the Instituto Nacional de Pesquisas da Amazônia (INPA).

**Author Contributions:** A.d.L.B., E.J.S.P.d.L., J.V.F. and L.R.D.A.: Conceptualization, Investigation, Methodology, Data curation, Writing—original draft. E.S.L., D.P.B., E.R.S., B.R.d.L., E.V.C. and M.L.B.P.: Data curation and Investigation. F.M.A.d.S., N.M.D.C., J.F.C.G.: Writing—review and editing. G.A.B. and H.H.F.K.: Writing—review and editing, Supervision. All authors have read and agreed to the published version of the manuscript.

**Funding:** We would like to thank Fundação de Amparo à Pesquisa do Estado do Amazonas (FAPEAM) for the funding of this research under the Universal Call FAPEAM-006/2019. The authors would also like to thank Conselho Nacional de Desenvolvimento Científico e Tecnológico (CNPq) (No. 305942/2020-4). This study was financed in part by the Coordenação de Aperfeiçoamento de Pessoal de Nível Superior—Brasil (CAPES)—Finance Code 001.

**Data Availability Statement:** Not applicable.

**Conflicts of Interest:** The authors declare no conflict of interest.

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
