# Peer review of "Cytotoxicity and Lipase Inhibition of Essential Oils from Amazon Annonaceae Species"

_chemistry, doi:10.3390/chemistry4040081_

Round 1
Reviewer 1 Report
The manuscript contains very interesting research on the biological activity of compounds obtained from Amazon Annonaceae Species. The manuscript fits perfectly into the contemporary trends related to the use of compounds obtained from plant material in the treatment of diseases affecting people, especially neoplastic diseases. The research methods and description of the obtained results are mostly sufficient, but a careful reading of the mancript brings some remarks:
- why in the extraction studies, for comparison, studies with a different solvent were not carried out, and another extraction method was not used?
- the manuscript lacks photos of the studied plants, I think it would significantly enrich the manuscript,
- is there information in the scientific literature about the toxic effects of the compounds tested in this manuscript?
- conclusions should be expanded, they should better summarize the results presented in the manuscript.
Author Response
Reviewer: 1
“Why in the extraction studies, for comparison, studies with a different solvent were not carried out, and another extraction method was not used?”
Response: Dear reviewer, in fact, the extraction of essential oils from barks of Duguetia flagelaris and Xylopia benthamii was carried out by hydrodistillation, which use water and is the most common method worldwide. The function of dichloromethane is just to wash the walls of the clevenger and thus collect oil droplets that were dispersed during extraction.
“The manuscript lacks photos of the studied plants, I think it would significantly enrich the manuscript.”
Response: We added in the supplementary material exsiccate images of the species whose essential oils were investigated.
“Is there information in the scientific literature about the toxic effects of the compounds tested in this manuscript?”
Response: Unfortunately, we did not find information in the literature about the toxic effects of the supposed active compounds.
“Conclusions should be expanded, they should better summarize the results presented in the manuscript.”
Response: The conclusion was improved as requested.

Reviewer 2 Report
The research article under the title “Cytotoxicity and Lipase Inhibition of Essential Oils from Am-azon Annonaceae Species” by Barros and coworkers presents interesting results on the study chemical composition of essential oils from Amazon species. The chemical profile of different species has been determined by the GC-MS, followed by the cytotoxic and lipase enzyme inhibition assessment. The most important compounds for the mentioned activities were obtained by PCA analysis and docked into selected proteins. My recommendation is MAJOR REVISION and the article will be suitable for publication after these questions are addressed by the authors:
1. Line 26 – the authors should correct the term “stablish”
2. Line 106 – which ultrapure chemical was used?
3. Line 208 – why is Elemol written with the capital letter E?
4. Line 233 – there is a space between l and ines
5. Line 261 – there is a space between sug and gest
6. Section 3.3. – the binding energies should be presented in kJ/mol
7. The authors should include the following reference for the importance of molecular docking in cytotoxicity studies:
D. S. Dimić, G. N. KaluÄ‘erović, E. H. Avdović, D. A. Milenković, M. N. Živanović, I. Potocnak, E. Samol’ova, M. S. Dimitrijević, L. Saso, Z. S. Marković, J. M. Dimitrić Marković, Synthesis, Crystallographic, Quantum Chemical, Antitumor, and Molecular Docking/Dynamic Studies of 4-Hydroxycoumarin-Neurotransmitter Derivatives, Int. J. Mol. Sci., 23, 1001, 2022. DOI: /10.3390/ijms23021001
8. The author should specify the groups of doxorubicin that are responsible for the interactions with amino acids, as well as functional groups of moieties in the investigated substances that form interactions with amino acids. Based on the structural considerations there should be a rationale for the importance of specific groups in these interactions. The same applies to all three proteins.
9. The conclusion should be rewritten to include the main quantitative data, as this way it is mostly discussion that is repeated.
Author Response
Reviewer: 2
“Line 26 – the authors should correct the term “stablish”
Response: It was changed as suggested.
“Line 106 – which ultrapure chemical was used?”
Response: The water used in this work was ultrapure grade. This information was added to the manuscript.
“Line 208 – why is Elemol written with the capital letter E?”
Response: Dear reviewer, thank you for noticing. “Elemol” as well as “δ-Cadinene” were capitalized by mistake. This information is now corrected in the manuscript.
“Line 233 – there is a space between l and ines”
Response: It was changed as suggested.
“Line 261 – there is a space between sug and gest”
Response: It was changed as suggested.
“Section 3.3. – the binding energies should be presented in kJ/mol”
Response: Dear reviewer, in order to be able to compare our results with other articles published on the same topic, we chose to keep the binding energies in kcal/mol. In this way, we are also in line with the recommendations presented in the largest review articles about molecular docking. For example, Pagadala et al., 2017 published a review article (more than 860 citations) entitled “Software for molecular docking: a review” in Biophysical reviews journal and said “…an affinity scoring function, ΔG [U total in kcal/mol], is employed to rank the candidate poses as the sum of the electrostatic and van der Waals energies.” New docking algorithm also describes the binding energies in term of kcal/mol. For example, Thomsen and Christensen published an article (more than 2000 citations) entitled “MolDock: A New Technique for High-Accuracy Molecular Docking” in Journal of Medicinal Chemistry and the binding energies were presented in kcal/mol. Finally, in the Autodock vina software”, which have almost 21000 citations, the predicted binding affinity is always presented in kcal/mol.
“The authors should include the following reference for the importance of molecular docking in cytotoxicity studies: D. S. Dimić, G. N. KaluÄ‘erović, E. H. Avdović, D. A. Milenković, M. N. Živanović, I. Potocnak, E. Samol’ova, M. S. Dimitrijević, L. Saso, Z. S. Marković, J. M. Dimitrić Marković, Synthesis, Crystallographic, Quantum Chemical, Antitumor, and Molecular Docking/Dynamic Studies of 4-Hydroxycoumarin-Neurotransmitter Derivatives, Int. J. Mol. Sci., 23, 1001, 2022. DOI: /10.3390/ijms23021001”
Response: We added in the introduction a sentence about computational approaches related to cancer and diabetes. The suggested reference was included in this sentence.
“The author should specify the groups of doxorubicin that are responsible for the interactions with amino acids, as well as functional groups of moieties in the investigated substances that form interactions with amino acids. Based on the structural considerations there should be a rationale for the importance of specific groups in these interactions. The same applies to all three proteins.”
Response: We improved the data presentation.
“The conclusion should be rewritten to include the main quantitative data, as this way it is mostly discussion that is repeated.”
Response: The conclusion was improved as requested.

Round 2
Reviewer 2 Report
The authors have answered all of the queries by the Reviewer.